# The Role of Tableware Size in Healthy Eating—Effects on Downstream Food Intake

**DOI:** 10.3390/foods12061230

**Published:** 2023-03-13

**Authors:** Sashie Abeywickrema, Mei Peng

**Affiliations:** 1Sensory Neuroscience Laboratory, Department of Food Science, University of Otago, Dunedin 9016, New Zealand; 2Riddet Institute, Private Bag 11 222, Palmerston North 4442, New Zealand

**Keywords:** tableware size, plate size, portion size perception, satiety, subsequent energy intake, obesity

## Abstract

Recent studies show that visual exposure to different portion sizes can lead to portion alterations in subsequent meals, suggesting that manipulations of tableware sizes may also modify portion size perception and downstream eating behaviour. The present study aims to address this novel question by testing 61 male participants (20–40 years; 19.7–41.5 kg·m^−2^) over three breakfast sessions in a controlled laboratory. In each session, the participant was served a pre-determined breakfast portion in either medium (control; CT), small (SC), or large (LC) jars. Participants were asked to rate post-meal satiety, and then recorded food intake for the rest of the day using Food Records. Our results indicated significant changes in post-meal satiety following the SC or LC condition, compared to CT (SC: 55.3 ± 10.8, LC: 31.0 ± 8.4, CT: 42.1 ± 9.6, F_(2, 108)_ = 25.22, *p* < 0.001). SC led to a reduction in post-breakfast energy intake (F_(2, 108)_ = 61.28, *p* < 0.001), but was counteracted by a substantial increase in downstream intake at the following meal (F_(2, 108)_ = 47.79, *p* < 0.001), resulting in an overall increase in total daily energy intake (F_(2, 108)_ = 11.45, *p* < 0.001). This study provides the first evidence that small tableware may not be a long-term solution for addressing overeating and related health issues (e.g., obesity), reinforcing the importance of considering downstream intake in eating-related intervention.

## 1. Introduction

The modern food environment, providing easy access to abundant supply, is thought to be one of the key contributors to a wide range of health issues, including obesity [1,2]. Over the past few decades, researchers from diverse disciplines have endeavoured to devise interventions to rectify the common phenomenon of overeating [3,4,5]. One promising approach is to manipulate environmental factors, such as lighting [6,7], sound [8,9], and tableware dimensions [10,11], to alter food perception and correspondingly facilitate healthier eating. However, to date, most studies have solely focused on assessing these effects for within-meal behaviour, and rarely consider food intake subsequent to the test meal—termed ‘downstream food consumption’ [12] in this paper.

In the fields of sensory nutrition and food psychology, there is increasing evidence for the importance of physical eating environments on food perception and consumption behaviour. Of the various factors that have been tested, tableware size is thought to play an important and direct role in activating implicit consumption norms, as many consumers use this visual cue to calibrate their food intake [10,11]. In keeping with the Delboeuf illusion, a small spatial ratio between food and tableware is hypothesised to give a perception of a more-than-actual portion, and thus lead to increased expected satiety [13,14,15]. This effect is particularly relevant in the context of pre-meal planning, when consumers determine their portion sizes, especially in cultures using individual-serving models, see Peng et al. [16]. 

While the approach based on tableware sizes is supported for its promise to reduce energy intake and food waste, and is being recommended by public health sectors [17], its efficacy has constantly been a subject of debate [10,18]. Indeed, studies using ad libitum serving models have found a mixture of results, with some detecting significant differences in intake [13,14,19] and others not [20,21,22]. In a systematic review of tableware effects [23], several possible explanations were put forward for the observed inconsistencies across previous studies. Specifically, these authors argued that the effects of tableware sizes are subject to distraction factors, types of containers, serving models, and types of food in some cases. A more recent review proposed that insufficient power was also likely an issue, pointing out that most existing studies had included too few participants [24]. 

In addition to methodological factors, studies of tableware sizes with different sub-populations appear to suggest that personal factors can substantially affect results. For instance, Peng, et al. [16] found that Asian consumers (e.g., Chinese and South Koreans) were affected by tableware sizes to a lesser degree than Western consumers (e.g., Canadians and New Zealanders), highlighting that cultural background appears to be a moderating factor of the tableware size effect. Shimpo and Akamatsu [25] later confirmed similar cultural differences using a Japanese cohort. More intriguingly, people from different weight groups possibly have different levels of susceptibility to tableware size manipulations [26]. Specifically, Peng [27] compared healthy-weight individuals versus overweight for their estimated intake of food presented on large versus small plates. Their results showed that overweight individuals were more likely to change their intake estimates in response to variations in tableware sizes. By contrast, Shah et al. [21], in a preliminary study, found no such differences among individuals in different weight groups. Broader research with regards to food-related visual biases has also produced inconsistent results regarding weight-specific differentiations, with some observing differences across weight groups in responding to food cues [28,29,30], while others did not [31,32,33]. Given the potential application of the effects of tableware size to improve healthy portion selection, particularly among overweight populations, more research is warranted to enhance our understanding of these effects and their interactions with weight groups.

Previous studies of tableware size effects have predominantly focused on consumption within a meal episode [13,34]. However, it remains unknown whether tableware size has effects on intake behaviour subsequent to the meal of interest. Studies of portion size effects have consistently suggested an accordance between portion size and energy intake within a meal—with larger portions almost always leading to increased intake [31,35,36]. Recently, Robinson et al. [37] found that small portions do not only have an impact on the intake within a meal episode, but also can lead to substantial reductions in energy intake on the subsequent day. Robinson et al. [36,37] further demonstrated that such long-lasting effects of small portions were not a mere learning outcome of post-ingestive behaviour, but rather the consequence of shifting portion size normality. According to these authors, simple visual exposure to small portions can lead the participants to reduce their choice of portion size for the subsequent meals. These recent findings point to the intriguing possibility that tableware size manipulations may also modify subsequent eating behaviour. 

The present study aims to test for tableware size effects on downstream food consumption subsequent to the testing meal. Building upon previous research exploring the portion size effect, we hypothesise that changes in tableware size may impact downstream food intake. More specifically, we predict that eating from smaller or larger tableware can lead to a shift of portion norm and thus increase or decrease downstream food intake. Overall, this study contributes to the longstanding debate regarding the efficacy of changing visual perception of food portion sizes via tableware size manipulation. The findings from this study can give important and timely implications for achieving more sustainable and healthy dietary behaviours.

## 2. Materials and Methods

### 2.1. Subjects

Males aged 20–40 years from the general community of Dunedin (New Zealand) were invited to participate in this study, initially advertised as a consumer food study. Only male participants were included to eliminate sex-related differences in visual perception. Individuals with chronic sensory dysfunction, neurological disease, or dietary restrictions (due to, e.g., allergies, religious practices, medications) or having body-mass-index (BMI) under 18.5 kg·m^−2^ were excluded from the study. Eligible participants were asked for their height (cm) and weight (kg) to guide representativeness of different weight groups [38]. A total of 61 participants participated in the study. 

Sample size was determined based on previously reported effect sizes of satiety and energy intake measures for tableware or portion size manipulations [27] using the G*Power 3.1.9.7 software [39]. Analyses suggested that a minimum of 45 participants would be sufficient to detect a medium-sized effect of jar size on satiety measures with a 90% power and an α-level of 0.05 using ANOVA: repeated measures, within between interactions (effect size *f* = 0.25). Correlation among repeated measures was 0.5, and non-sphericity correction was 1. We recruited an additional 16 participants to adjust for attrition or missing data (e.g., physiologically implausible food records) [40,41]. 

Informed written consent was obtained from each participant prior to the study. Participants were given monetary compensation upon completion. The study was approved by the University of Otago Human Ethics Committee (Reference: 20/108).

### 2.2. Testing Food Models

Oat is one of the common breakfast choices among New Zealand consumers [42]. Notably, data from dietary records collected for our previous studies (N > 400) showed that oat pudding is the most familiar breakfast to local consumers. A formulation of water-based overnight oat pudding was, therefore, specially developed for this study, using rolled oats (Harraway and Sons Ltd., Dunedin, New Zealand), chia seeds (Alison’s Pantry, Hamilton, New Zealand), and carbon-filtered water (rolled oat: chia seeds: filtered water ratio = 1 g:0.07 g:2.28 mL). It derives approximately 464 kJ per 100 g, consisting of 16.8 g carbohydrate, 3.6 g protein, and 2.6 g fat. 

### 2.3. Serving Jars

Three types of glass jars with the same height (11.5 cm) but varying diameters were used in the small container (SC; d = 5.7 cm; 9.9 oz), large container (LC; d = 8.9 cm; 24.2 oz), and control (CT; d = 7.3 cm; 16.3 oz) conditions (Figure 1). All the glass jars were from the same supplier (Arthur Holmes, Petone, New Zealand). Figure 1 shows three glass jars in one frame for comparison; however, no study participants were shown them displayed together.

### 2.4. Experimental Procedure 

Each participant of the study attended four morning sessions, following a >10 h overnight fasting (either at 0700–0730 h, 0745–0815 h, 0830–0900 h, or 0915–0945 h), at the Sensory Neuroscience Laboratory, University of Otago. Any two sessions were at least two days apart, with each participant’s starting weekday being randomised to mitigate behavioural biases. The four sessions included an initial session and three testing sessions. Orders of the three testing sessions were randomised across the participants following a William Latin Square design [43]. Participants maintained their exercise levels across testing days.

Upon arrival at each session, the participants were asked to rate the level of hunger and fullness on a Satiety Labeled Intensity Magnitude (SLIM) scale (Anchors of the SLIM rating include; greatest imaginable hunger  =  −100.0; extremely hungry  =  −67.4; very hungry  =  −56.2; moderately hungry  =  −38.2; slightly hungry  =  −18.6; neither hungry nor full  =  0; slightly full = 31.9; moderately full  =  46.7; very full  =  74.3; extremely full  =  79.4; greatest imaginable fullness  =  100): [44,45]. 

In the initial session, the participant was presented with five glass jars (all in medium jar), which contained 159 g to 388 g oat pudding following a logarithmic scale with a step of 0.1. The participants were asked to rate each sample for expected satiety on a 100-point Visual Analogue Scale (VAS; 0 = not full at all; 100 = extremely full): [46], and then to select a sample as ideal portion size for breakfast. This portion was then used as their self-selected portion size in the subsequent sessions.

In each of the next three testing sessions, the participant was served oat pudding of their self-selected portion sizes in medium (CT), small (SC), or large jars (LC). Notably, 7 out of the 61 participants selected portions that exceeded the volume of the SC jar size, which was not anticipated. These participants were served a full SC jar (i.e., 287 g) and an additional portion in a plastic portion cup. After the consumption task in each session, the participant was asked to report post-meal satiety on a SLIM scale and hedonic response on a 100-point VAS (anchors; 0 = very unpleasant; 100 = very pleasant). 

After each breakfast session, the participants were required to record all foods and beverages consumed within a day, following a standard 24 h weighed Food Record [12,47]. This Food Record was proofed by a NZ registered dietitian and nutritionist. An electronic food scale (Model No. 1023, Salter, Manchester, UK) was provided, along with a food portion catalogue containing imagery measures of portion sizes (e.g., for dining out). Additionally, the participants reported alcohol, supplements, medicines intake, and any event that might influence their eating behaviour. The same experimenter delivered instructions to all participants.

All the participants were then requested to complete a Dutch Eating Behaviour Questionnaire (DEBQ); [48] and a demographic questionnaire, including questions regarding their physical activity (i.e., bed rest to very heavy/vigorous activity; Capra) [49]. At the end of the last session, each participant’s height and weight were measured in light clothing without shoes. 

Finally, each participant was asked to write down their thoughts on the study aim and differences of the oat pudding across sessions (Q1: Please use the space below to write down what you think the study was about; Q2: Please write down if you have noticed any sensorial difference about the oat puddings across the three sessions). The purpose of the study was not disclosed to the participants until the completion of their participation. The participant was then given the opportunity to withdraw data. 

### 2.5. Data Analyses

#### 2.5.1. Data Pre-Processing

Individual weighed Food Records of each experimental session were separately entered into FoodWorks (Brisbane, Australia: Xyris Pty Ltd., 2019), which translated individual food consumption into energy intake (in kJ). We applied the method of Huang et al. [50] to identify physiologically implausible dietary reports, which prescribes calculations of individual predicted energy requirement for the reporting period (pER; via Harris–Benedict equation; Roza and Shizgal [51], and self-reported physical activity level). Using ±1.5 SD cut-offs, individuals whose reported energy intakes (rEI) outside of 67–133% over pER were considered implausible. With this method, 19 reports were identified to be under-reported and 2 were over-reported. This misreporting rate (i.e., 35%) was in line with previous studies [40,41]. 

Averaged daily food intake (in kJ) from participants with physiologically plausible Food Records (N = 40) was extracted for analyses. In addition, the participant’s food intake within five-time intervals, including post-breakfast, lunch, post-lunch, dinner, and post-dinner, were extracted (in kJ). Time gaps between two adjacent meals (e.g., breakfast–lunch) were approximately 4-to−6 h. In addition, individual BMI was calculated as kg/m^2^, where kg is the participant’s weight in kilograms and m^2^ is their height in metres squared.

#### 2.5.2. Statistical Analyses

The main statistical analyses of the study were pre-specified and shared at https://osf.io/rdb4p/ (uploaded to OSF storage on 30 November 2021). However, some modifications were made during the publication process to give more insights. Specifically, originally proposed univariate comparisons across BMI groups were changed to using BMI and self-selected Portion Size as continuous covariates in the main analyses. Additionally, energy intakes for separate time intervals were separately evaluated with additional repeated-measures analysis of covariance (ANCOVA). Lastly, an additional analysis of order effects was performed. 

Participant characteristics were summarised with descriptive statistics (e.g., age, BMI), with additional Cronbach’s alpha coefficients calculated for each DEBQ subscale to indicate internal consistency. ANCOVAs were employed to test differences for baseline SLIM and hedonic ratings across the Conditions, with BMI being treated as a covariate. Generalised Linear Mixed Models ANOVA was used to test whether the order of the experimental session had any effect on the primary outcome variables. 

For the main analyses, repeated-measures ANCOVAs were separately applied to analyse the two primary testing outcomes—i.e., post-meal SLIM ratings (for initial and post-processing data; N = 61 or 40) and total daily energy intake obtained from the Food Records (for post-processing data only; N = 40). In each analysis, the *Condition* (i.e., CT, SC, LC) was treated as the within-subject variable, with *BMI* and the self-selected *Portion Size* of the test meal as continuous covariates. Any significant effect was explained by post-hoc tests with *Bonferroni* multiple comparisons [52]. 

In order to give more insights into the observed effects against time, energy intake within five-time intervals was extracted from the Food Records, including post-breakfast, lunch, post-lunch, dinner, and post-dinner. Separate repeated-measures ANCOVAs were used to assess the differences across *Conditions*, while controlling for *BMI* and self-selected *Portion Size*. 

Significance was indicated by *p* < 0.05. All the statistical analyses were performed using R-software (version 1.1.463, RStudio, Boston, MA, USA). 

## 3. Results

### 3.1. Participant Characteristics

Table 1 summarises participant characteristics and baseline SLIM/hedonic measures for all participants (N = 61) and for the post-processing dataset (N = 40). Overall, the BMIs of the participants were 28.3 ± 6.3 kg∙m^−2^, comparable to national health reports [53,54]. Cronbach’s alpha coefficients for DEBQ restrained, emotional, and external subscales were 0.79, 0.93, and 0.80, respectively, all exceeding the criterion of 0.70 for internal consistency [55]. The summary of 40 participants did not vary substantially from the overall statistics. The main datasets can be found at https://osf.io/3uecx/ (data has been deposited on 6 April 2022).

No significant differences were found across *Conditions* in terms of baseline SLIM rating (F_(2, 177)_ = 1.34, *p* = 0.541, *ηp*^2^ = 0.01) or hedonic ratings (F_(2, 177)_ = 1.74, *p* = 0.243, *ηp*^2^ < 0.01). BMI was not a significant covariate in either model (*p* > 0.05).

### 3.2. Effects of the Session Order

The Generalised Linear Mixed Model ANOVAs on post-meal SLIM ratings and total daily energy intake did not detect a significant interaction between *Condition* and *Session Order* (post-meal SLIM ratings: F_(4, 108)_ = 1.17, *p* = 0.270, *ηp*^2^ = 0.06; total daily energy intake: F_(4, 108)_ = 0.12, *p* = 0.902, *ηp*^2^ = 0.04). 

### 3.3. Comparisons of Within-Meal Measures across Three Jar Size Conditions

With regard to post-meal SLIM ratings, analysis based on 61 participants showed a significant main effect due to *Condition* (F_(2, 171)_ = 9.01, *p* = 0.010, *ηp*^2^ = 0.22, Figure 2A). Post-hoc tests with *Bonferroni* corrections revealed that the SC condition yielded significantly higher SLIM ratings than CT (t_(114)_ = −3.40, *p* = 0.010, *d* = −0.44) or LC (t_(114)_ = −5.33, *p* = 0.009, *d* = −0.76), with the latter being the lowest (t_(114)_ = 3.97, *p* = 0.005, *d* = 0.66). Neither *BMI* nor *Portion Size* was a significant covariate (*BMI*: F_(1, 171)_ = 1.09, *p* = 0.330, *ηp*^2^ = 0.02, *Portion Size*: F_(1, 171)_ = 2.60, *p* = 0.104, *ηp*^2^ = 0.04). 

The analyses based on post-processing data (N = 40) revealed similar results, with *Condition* having a significant effect on post-meal SLIM ratings (F_(2, 108)_ = 25.22, *p* < 0.001, *ηp*^2^ = 0.45; Figure 2B). SC yielded a significantly higher SLIM rating than LC (t_(72)_ = −7.52, *p* < 0.001, *d* =−1.47) or CT conditions (t_(72)_ = −3.35, *p* = 0.006, *d* = −0.58), with LC being lower than CT (t_(72)_ = 4.17, *p* < 0.001, *d* = 0.75). Moreover, neither *BMI* nor *Portion Size* was a significant covariate (*BMI*: F_(1, 108)_ = 2.04, *p* = 0.156, *ηp*^2^ = 0.04, *Portion Size*: F_(1, 108)_ = 3.27, *p* = 0.075, *ηp*^2^ = 0.07).

### 3.4. Effects of Jar Size on Subsequent Energy Intake

ANCOVA on the total daily energy intake of 40 participants revealed a significant main effect due to *Condition* (F_(2, 108)_ = 11.45, *p* < 0.001, *ηp*^2^ = 0.42). Post-hoc tests indicated that SC (10,181 ± 2777 kJ) led to a significantly higher total daily energy intake than LC (9346 ± 2391 kJ; t_(72)_ = −5.15, *p* < 0.001, *d* = −0.67) and CT conditions (8674 ± 1715 kJ; t_(72)_ = −4.10, *p* < 0.001, *d* = −0.54), with no significant difference between the latter pair (t_(72)_ = 1.99, *p* = 0.404, *d* = −0.08). Neither *BMI* nor *Portion Size* was a significant covariate (*BMI*: F_(1, 108)_ = 2.58, *p* = 0.174, *ηp*^2^ = 0.06, *Portion Size*: F_(1, 108)_ = 2.80, *p* = 0.139, *ηp*^2^ = 0.05). 

Furthermore, individual differences in total daily energy intake across *Condition* were assessed against pER. Results indicated that, on average, the difference in total daily energy intake between SC and CT was 2407 ± 404 kJ, representing approximately 23% of pER. The energy intake difference between SC and LC was 1734 ± 320 kJ, counting for 20% of pER. 

Separate repeated-measures ANCOVAs were performed on energy intake within different time intervals of the day (i.e., post-breakfast, lunch, post-lunch, dinner, post-dinner). The results revealed significant differences in post-breakfast (F_(2, 108)_ = 61.28, *p* < 0.001, *ηp*^2^ = 0.51) and lunch (F_(2, 108)_ = 47.79, *p* < 0.001, *ηp*^2^ = 0.45; Figure 3). Specifically, the SC and LC conditions significantly reduced and increased post-breakfast snack intake compared to CT (SC: t_(72)_ = 3.10, *p* = 0.007, *d* = −0.93; LC: t_(72)_ = −4.20, *p* < 0.001, *d* = 1.56; SC-LC: t_(72)_ = 4.13, *p* < 0.001, *d* = −2.17). Assessment of the lunch data suggested that the SC condition was associated with a significantly higher energy intake than the LC (t_(72)_ = −3.56, *p* < 0.001, *d* = 1.58) and CT conditions (t_(72)_ = −3.56, *p* < 0.001, *d* = 1.87). No significant difference was present at other eating episodes (post-lunch: F_(2, 108)_ = 0.77, *p* = 0.464, *ηp*^2^ = 0.01; dinner: F_(2, 108)_ = 1.01, *p* = 0.365, *ηp*^2^ = 0.02; post-dinner: F_(2, 108)_ = 1.90, *p* = 0.153, *ηp*^2^ = 0.03). *BMI* and *Portion Size* were not significant covariates (all *p* > 0.05).

### 3.5. Debrief from the Participants

Thirty-four out of forty participants believed that this study was to investigate consumer acceptance of different products of oat puddings (as advertised during the participant recruitment). Four other participants thought the study was to assess the health benefits of oats, while the remaining two participants did not provide an answer. With regards to differences across sessions, 24 participants mentioned that they noticed variable sizes of the breakfasts, along with other sensory differences (e.g., sweetness and texture). Notably, none of the participants mentioned differences in serving utensils (e.g., jars, spoons) in their answers. 

## 4. Discussion

Effects of tableware size on food consumption have represented a controversial research topic, characterised by conflicting findings [18,56]. The present study tested the effects of tableware (i.e., jar) size on satiety and downstream food consumption subsequent to the test meal. Our results showed that using small tableware can effectively lead to increased satiety. Furthermore, this study provides the first demonstration that tableware size can potentially alter downstream effects on energy consumption in subsequent meals. 

The present analysis of satiety measures indicated that serving food using large tableware significantly diminishes the feeling of satiety at the end of the meal. This finding is in line with the original hypothesis of the tableware size effect, and consistent with many previous studies that also measured satiety as a primary variable for detecting this effect [27,57]. Notably, many other studies of tableware size effects employ a double-serving, ad libitum method, which requires participants to self-serve from a large tableware to their immediate serving tableware. With this approach, tableware size effects on satiety are often offset, or even counteracted by participants having multiple servings [21]. Additionally, the present study did not find BMI to be a significant moderator of tableware size effects, with individuals of all BMI groups showing similar levels of responses to glass jar variations. These results were in line with Shah et al. [21]. 

Our study indicated that using small tableware led to increased overall energy intake. Moreover, temporal analyses indicated that small tableware initially led to reductions in energy intake but was followed by substantial increases in energy intake at the following meal. The immediate reduction in intake was a likely outcome of increased post-meal satiety when eating from small jars, in line with some previous reports [58], but see [22] for contradictory findings. The subsequently increased energy intake at lunchtime may be interpreted in two ways. First, the increased lunch intake can be seen as compensatory eating in response to the preceding low intake [36]. An alternative explanation relates to the theory put forward by Robinson et al. [37]—exposure to an altered portion size (either larger or smaller than the original norm) causes sustained shifts in perception of “normal” food portions, and thus affects the amount of food eaten in subsequent meal(s). Given that a constant portion size (for each participant) was used across our study, the observed differences in downstream energy intake reiterate the point made by Robinson et al. [36,37] that shifting portion norms do not necessarily involve post-ingestive behaviour, but rather may stem from visual exposure.

Findings from the present study have some important new implications with regard to eating interventions via tableware sizes. Despite highly controversial results, some health organisations have promoted the use of small tableware to help with portion reduction within a meal. The present findings cautioned against using tableware manipulations for eating interventions. Even though using smaller tableware appeared to have positive effects on reducing initial intake, it led to a substantial increase in daily energy intake (i.e., 2407 kJ, 23% of pER). From a broader perspective, downstream food consumption should be considered in assessments of food-related interventions.

The present study had a few limitations to consider. First, we used glass jars which are more prone to biases due to individual factors and task instructions, as opposed to plates/bowls [59,60]. Additionally, data from seven participants whose selected portion exceeded the small jar capacity led to additional uncountable variabilities. Another potential limitation of the current study relates to the use of self-reported Food Records. Despite representing the ‘gold standard’ in nutritional science for recording habitual energy intake [40], the self-reporting nature of this measure is prone to biases [41,61,62]. Another caveat of the current study is that it comprised a highly homogenous testing population (males aged between 20–40 years), which may limit the generalisability of our results. Future research is required to repeat our findings with a wide heterogeneous population. 

## 5. Conclusions

In summary, the present study reveals that using small tableware may lead to increased post-meal satiety and initial reductions in energy intake, but counteracted by substantial energy increase in the following meal, influencing sustainable dietary intake. These findings thus suggest that small tableware may not be a long-term solution for combating over-consumption. In more general terms, downstream effects of dietary intervention should be considered in future studies. Effective interventions for achieving a more sustainable and healthy diet should not only have within-meal impacts, but also have influences over a sustained period. 

## Figures and Tables

**Figure 1 foods-12-01230-f001:**
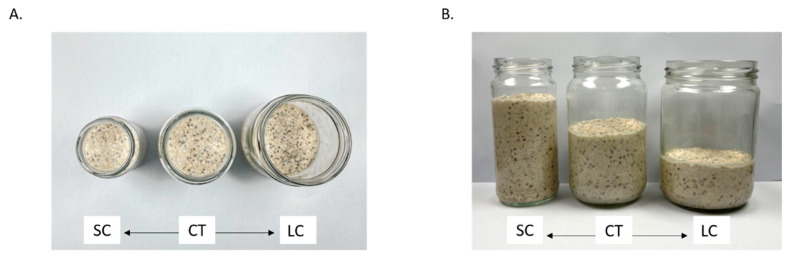
Demonstration of the top view (**A**) and the side view (**B**) of the jar sizes used in the study. Small jar size condition (SC) was 5.7 cm in diameter (volume = 9.9 oz), medium jar size condition (control, CT) was 7.3 cm in diameter (volume = 16.3 oz), and large jar size condition (LC) was 8.9 cm in diameter (volume = 24.2 oz). The three jars displayed in the figure contained 226 g of oat pudding—a portion averaged across all participants.

**Figure 2 foods-12-01230-f002:**
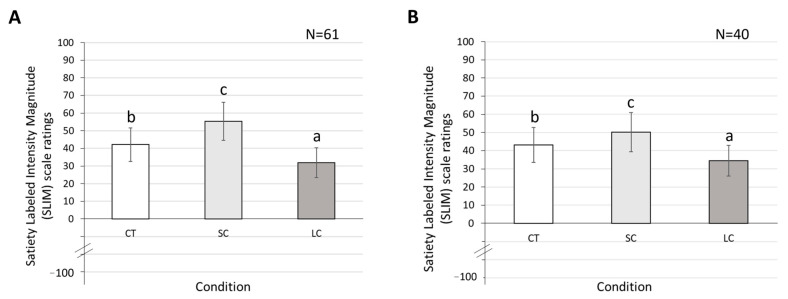
Effect of the jar size condition (CT, SC, LC) on Satiety Labelled Intensity Magnitude (SLIM) scale ratings based on 61 (panel (**A**)) or 40 participants (panel (**B**)). Error bars represent standard deviations. Assigned letters in the figure indicate significant differences (*p* < 0.05) across the conditions.

**Figure 3 foods-12-01230-f003:**
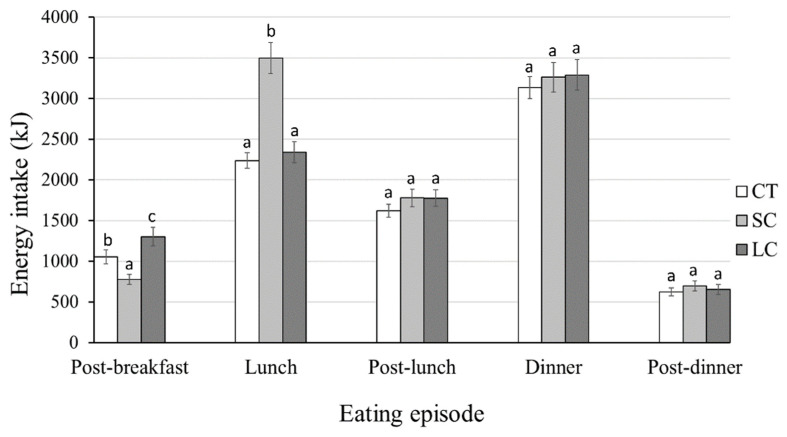
Bar charts illustrating the downstream energy intake across jar size conditions (CT, SC, LC) at post-breakfast, lunch, post-lunch, dinner, and post-dinner eating episodes for N = 40 participants. Data are reported as mean ± standard error. Assigned letters in the figure indicate significant differences across the conditions at each eating episode (*p* < 0.05).

**Table 1 foods-12-01230-t001:** Summary of participant characteristics, self-selected portion size (g), and baseline Satiety Labeled Intensity Magnitude (SLIM) ratings across the three experimental sessions. Baseline SLIM ratings are reported for each small jar size condition (SC), medium jar size condition (control, CT), and large jar size condition (LC).

	Participants Recruited into the Study (N = 61)	Participants Included in the Main Analyses (N = 40)
	Mean ± Standard Deviation (Range)	Mean ± Standard Deviation (Range)
**Age (years)**	28 ± 7 (20–40)	27 ± 6 (21–39)
**BMI (kg∙m^−2^)**	28.3 ± 6.3 (19.7–41.5)	27.7 ± 5.9 (19.7–41.5)
**DEBQ eating score** Restrained Emotional External	2.2 ± 0.4 (0.9–3.5) 2.2 ± 0.8 (1.0–4.8) 3.4 ± 0.7 (1.4–4.5)	2.0 ± 0.6 (1.0–3.3) 2.1 ± 0.8 (1.0–4.7) 3.2 ± 0.6 (1.9–4.3)
**Self-selected portion size (g)**	259.8 ± 65.3 (159.0–388.0)	226.0 ± 46.2 (159.0–310.4)
**Baseline SLIM ratings** CT SC LC	−58.3 ± −17.8 (−19.8 to −72.3) −60.5 ± −22.4 (−17.6 to −83.4) −64.5 ± −20.0 (−20.6 to −72.1)	−56.8 ± −19.5 (−23.1 to −70.2) −59.4 ± −21.6 (−19.4 to −83.4) −60.3 ± −17.9 (−25.8 to −65.4)
**Hedonic VAS ratings** CT SC LC	31.5 ± 22.2 (4.1 to 57.0) 32.4 ± 25.4 (3.8 to 58.8) 31.7 ± 20.8 (4.3 to 58.1)	30.6 ± 18.5 (5.3 to 52.2) 29.5 ± 21.0 (3.8 to 55.7) 31.2 ± 19.7 (5.1 to 57.0)

## Data Availability

Data have been deposited in repository in Open Science Framework, OSF; https://osf.io/3uecx/ (data has been deposited on 6 April 2022).

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
