# Peer review of "The Role of Tableware Size in Healthy Eating—Effects on Downstream Food Intake"

_foods, 2023, doi:10.3390/foods12061230_

Round 1
Reviewer 1 Report
Topic of your research is very interesting. Please follow instructions given in revised word file.

Author Response
Please find the attached response letter.

Reviewer 2 Report
This manuscript entitled “The role of tableware size in healthy eating - effects on downstream food intake.” is an interesting and original study.
The paper is clearly presented and results are very useful. However, it should be completed because it is too short for a major journal such as Foods. Please expand it with some more analysis and improve it according to some suggestions.
1. Text and tables. The number of digits of the error value depends on the place where the significant digit appears and the number of digits of the corresponding data should be adjusted by taking into account the corresponding error values. In this way, each value in the Tables must be expressed with the significant digits according to the significant digits of each error value (in this case the significant digits of the standard deviation). Please correct it.
2. Put all units in the international system
Author Response

(The authors gave the same response as above.)
